# MARS: Toward More Efficient Multi-Agent Collaboration for LLM Reasoning

## Abstract

Large language models (LLMs) have achieved impressive results in natural language understanding, yet their reasoning capabilities remain limited when operating as single agents. Multi-Agent Debate (MAD) has been proposed to address this limitation by enabling collaborative reasoning among multiple models in a round-table debate manner. While effective, MAD introduces substantial computational overhead due to the number of agents involved and the frequent communication required. In this paper, we propose MARS (Multi-Agent Review System), a role-based collaboration framework inspired by the review process. In MARS, an author agent generates an initial solution, reviewer agents provide decisions and comments independently, and a meta-reviewer integrates the feedback to make the final decision and guide further revision. This design enhances reasoning quality while avoiding costly reviewer-to-reviewer interactions, thereby controlling token consumption and inference time. We compared MARS with both MAD and other state-of-the-art reasoning strategies across multiple benchmarks. Extensive experiments with different LLMs show that MARS matches the accuracy of MAD while reducing both token usage and inference time by approximately 50%. Code is available at `https://anonymous.4open.science/r/ICLR2026-submit-F7B0/README.md`.

## 1 Introduction

Recent advances in large language models (LLMs) have brought revolutionary changes to the field of natural language processing and understanding (Minaee et al., 2024; Hadi et al., 2023; Li et al., 2025; Brown et al., 2020). With their unprecedented performance on language generation tasks such as chat completion, question answering, and code generation, these models are expected to extend their impact to more general and challenging real-world applications, including knowledge-intensive reasoning. However, when it comes to problems involving deep factual knowledge and complex structures, their reasoning abilities still remain limited (Mirzadeh et al., 2024; Jiang et al., 2024b). For instance, many state-of-the-art models can only achieve about 30% - 40% accuracy on the GPQA benchmark.

To enhance reasoning during inference, several prompting techniques have been proposed, including Chain-of-Thought (CoT) (Wei et al., 2022), Tree-of-Thought (ToT) (Yao et al., 2023), and self-reflection (Shinn et al., 2023). While these approaches significantly improve LLM performance, they operate in a single-agent setting, where the model lacks external feedback and often fails to correct its own mistakes. Motivated by this limitation, recent works have explored multi-agent collaboration, in which agents provide feedback to each other to enhance reasoning. Among these, the Multi-Agent Debate (MAD) framework (Du et al., 2023) is the most widely adopted: multiple agents are recruited to tackle the same problem, exchange responses, and refine their answers by considering disagreements, with the final solution derived via majority voting after a round-table discussion.

Despite MAD achieving a significantly improved performance in reasoning, a key drawback is its high resource cost: both token consumption and inference time increase substantially compared to single-agent systems, due to intensive communication among agents. This raises a fundamental question: **Can we reduce resource consumption while preserving the response quality of multi-agent reasoning?**

To address this challenge, we draw inspiration from two sources: verifier-based architectures and the hierarchical evaluation structure of peer review. On one hand, recent studies on agent tuning with synthesized trajectories incorporate verifiers or critics as key components (Fu et al., 2025; Gao et al., 2024; Shi et al., 2025). These verifiers—rule-based or LLM-based—check and filter low-quality trajectories or intermediate results, underscoring the importance of systematic review and quality control. On the other hand, peer review practices demonstrate the effectiveness of hierarchical evaluation: authors (or agents) refine their work based on structured feedback, while final decisions are reached through systematic aggregation. Unlike the round-table discussion in MAD, where agents communicate freely, peer review emphasizes independence—reviewers provide evaluations without direct interaction—yet the process reliably converges to fair and consistent outcomes.

Based on these considerations, we propose **M**ulti-**A**gent **R**eview **S**ystem (**MARS**) for LLM reasoning, in which agents take on the roles of author, reviewers, and meta-reviewer to mimic a structured review process. Given a user query, the author agent first generates an initial response. Instead of producing their own answers from scratch, reviewer agents evaluate this response, focusing on identifying potential errors. Different to the round-table discussion in MAD, where agents engage in frequent communication, reviewers in MARS operate independently, each providing a decision, review comments, and a confidence score. The meta-reviewer then aggregates these evaluations and issues a final judgment with justification: if the decision is *accept*, the author's initial response is adopted as the final answer; if the decision is *reject*, the meta-reviewer generates feedback to guide the author's revision. By structuring collaboration around independent review and feedback, MARS matches the reasoning quality of MAD while consuming substantially fewer tokens and requiring less inference time.

We conducted a series of experiments on three commonly utilized reasoning benchmarks, using both closed-source LLMs (e.g., ChatGPT) and open-source models. With the same number of agents, MARS consistently reduced the resource usage while keeping the accuracy comparable to MAD across all datasets (Table 1). For example, the token consumption of ChatGPT on GPQA is reduced from 5042 to 2479 per query, while the inference time is reduced from 11.92 to 6.01 seconds. These results demonstrated both the effectiveness and efficacy of the MARS framework. In summary, this work has the following major contributions:

- We propose **MARS**, a role-based multi-agent framework by following a hierarchical evaluation process to increase the computational efficiency and preserve the performance of multi-agent reasoning.

- Experiments on multiple benchmarks validated that MARS matches the accuracy of the SOTA method MAD while reducing token usage and inference time by ∼50% when implementing with different LLMs.

- Through the development and benchmarking of MARS, we validated that adopting a mixture of different LLMs may affect the performance of reasoning.

## 2 RELATED WORK

**LLM Reasoning**. The reasoning capability of LLMs has become a research focus in recent years, with various methods proposed at both the training and inference stages. Prompting-based approaches, such as CoT(Wei et al., 2022), ToT(Yao et al., 2023), and self-reflection(Shinn et al., 2023), introduce structured pipelines to guide the model to decompose the problem and think step-by-step before generating the final answer. These methods are more straightforward and efficient, but the effectiveness is limited by the model's inherent reasoning ability. In parallel, there are also advanced reasoning models or agents which are finetuned with reasoning trajectories(Gao et al., 2024; Tan et al., 2025; Huang et al., 2022). These models come with reasoning and tool usage abilities naturally, but will require more computational resource for model training.

**Multi-agent Collaboration**. A single agent's capability is often insufficient for problems that require complex reasoning and specialized knowledge. Recent research also revealed that LLMs tend to adhere to their initial answers and struggle to identify and correct their own mistakes without external information. To address this, many studies have started to explore the collaborative behaviors of multiple agents. MAD is the most prominent framework in this line(Du et al., 2023; Smit et al., 2023; Liang et al., 2023), where multiple agents work on the same problem through round-table

discussion, iteratively exchanging responses and refining their answers by leveraging the feedback from others. While effective in improving reasoning, MAD introduces substantial computational overhead: both token consumption and inference time grow rapidly with the number of agents and communication rounds, making it costly for practical deployment.

**Answer Verification**. Many finetuning methods for LLM agents rely on synthetic reasoning and tool usage trajectories(Sun et al., 2025; Ficek et al., 2025; Lee et al., 2025; Fu et al., 2025). To ensure the quality of the structure, intermediate steps, and final results of the trajectories, recent works have designed various verifiers for quality control. These verifiers are either rule-based modules with human-defined principles, or LLM-based utilizing the inherent error-detection and evaluation abilities of large models. Unlike primary agents, verifiers are not required to generate full answers to user queries; instead, they specialize in detecting mistakes in reasoning steps and final results. The effectiveness of this design in improving reliability has been demonstrated in various agent systems. In our framework, reviewer agents serve a similar role: they provide independent decisions, comments, and suggestions, offering the author agent clearer and more explicit feedback to guide revisions of its initial response.

## 3 METHODOLOGY

### 3.1 OVERVIEW

In this section, we introduce the MARS framework, which is designed for reducing resource demands in multi-agent systems. We begin with the problem formulation, then describe the roles assigned to different agents, and finally outline the architecture and mechanisms of how the MARS framework works for complex reasoning tasks.

**Problem Formulation.** For a given reasoning task, let $x_i$ denote the input prompt (original problem description wrapped by our prompting templates), and let $y_i$ denote the model's response containing the final answer. Different from MAD where multiple agents are equivalent to each other, MARS adopts a role-based design. LLMs are employed to play three distinct roles, namely **author**, who generates initial responses, **reviewer**, who generates evaluation on the author's output, and **meta-reviewer**, who integrates reviewers' comments and provides feedback for the author. We use $\mathcal{A}$ to denote the author agent, $\mathcal{R}_j, j = 1, 2, ..., m$ to denote the reviewer agents, and $\mathcal{M}$ to denote the meta-reviewer agent, respectively.

### 3.2 MARS FRAMEWORK

The MARS framework implements a structured, role-based collaboration among agents to improve both reasoning accuracy and efficiency (Figure 1). Given a user query, the author agent generates an initial response. This output is then independently evaluated by multiple reviewer agents, each producing a decision, explanatory comments, and a confidence score without interacting with each other. A meta-reviewer agent aggregates these reviews to issue a judgment and, when necessary, provides targeted feedback to guide the author's revision. This hierarchical evaluation and feedback loop enables efficient parallel review while enhancing robustness through systematic error detection and correction. The details of each component and step are provided below (a summary of the procedures is available in Algorithm 1).

**Author's initial response.** For an input prompt $x_i$, the **author agent** generates an initial response containing both the reasoning trajectory and the final answer. To make the subsequent review process verifiable, we adopt the Chain-of-Thought (CoT) paradigm, requiring the author to output explicit intermediate reasoning steps alongside the final answer (see Appendix D.1.1 for the prompt template). Mathematically, this step can be presented as a function:

$$t_i, y_i = \mathcal{A}(x_i) \tag{1}$$

where $t_i$ denotes the sequence of reasoning steps and $y_i$ denotes the final answer.

**Reviewer's evaluation.** After the author produces an initial response, the review stage begins with $m$ **reviewer agents**, denoted $\mathcal{R}_1, \mathcal{R}_2, \ldots, \mathcal{R}_m$. Each reviewer receives the author's output as input and generates a review comprising: (i) a binary decision (accept or reject), (ii) a confidence score, and (iii) textual justification. If the decision is *reject*, the justification must explicitly identify errors

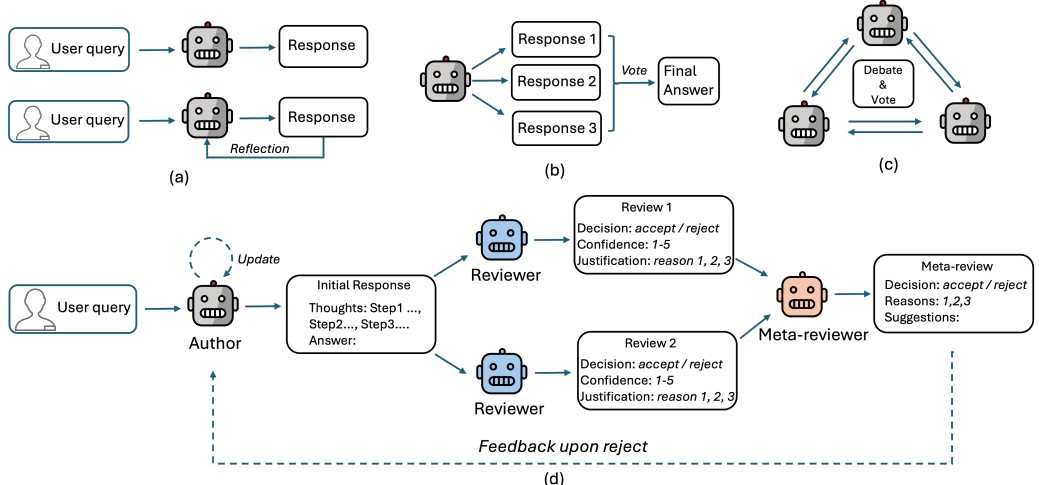

Figure 1: Overview of the architecture of MARS and baselines. (a) Basic single model inference (top) and self-reflection (bottom). (b) Self-consistency. (c) Multi-Agent-Debate (MAD). (d) Multi-Agent Review System (MARS). In MARS, the author agent receives a user query and generates an initial response. Each reviewer agent evaluates the response and provides a decision, confidence level, and justification (e.g., reasons for the decision, identified author mistakes). The meta-reviewer integrates review comments and makes the final decision, with suggestions for answer revision. Finally, the author agent incorporates the feedback and updates its response, leading to enhanced reasoning.

in either the intermediate reasoning steps or the final prediction (see Appendix D.1.2 for the prompt template). The review process of the $j$-th reviewer for the input $x_i$ can be mathematically presented as a function:

$$r_{ij} = \mathcal{R}_j(x_i, t_i, y_i), \quad j = 1, 2, ..., m \tag{2}$$

**Meta-review and feedback.** Unlike the normal peer review process, we do not incorporate direct feedback from the reviewers to the author after the review stage. Instead, a **meta-reviewer agent** consolidates reviewer feedback, makes decisions, and communicates with the author. This design is driven by two factors: (1) **Efficiency** — a direct discussion between reviewers and the author would substantially increase token usage and inference time, reducing reasoning efficiency; (2) **Role simplification** — Exposing the author to multiple, potentially redundant or conflicting comments would impose extra decision-making, such as filtering overlapping feedback or resolving disagreements, which could distract it from revising the response. We consider that by centralizing feedback integration in the meta-reviewer, the author remains focused on updating their answer, thereby maintaining reasoning quality.

The meta-reviewer processes all reviewer outputs, summarizes comments, resolves conflicts or redundancies, and issues a final decision. If the decision is *accepted*, the author's initial response is adopted as the final response. If the decision is *rejected*, the meta-reviewer provides explicit feedback to the author, including (i) the reasons for rejection (e.g., errors in the reasoning path) and (ii) concrete suggestions for improvement (see Appendix D.1.3 for the prompt template). This meta-reviewer can be mathematically presented as a function:

$$m_i = \mathcal{M}(x_i, t_i, y_i, r_i) \tag{3}$$

, where $m_i$ denotes the meta-reviewer's output, and $r_i$ is a concatenation of all the reviewer outputs, $r_i = r_{i1} \oplus r_{i2} \oplus ... \oplus r_{im}$

**Rebuttal stage.** After the meta-review stage, a final decision is issued regarding the author's initial response. If the decision is *rejected*, the author agent is required to revise the solution by leveraging both its reasoning history and the meta-reviewer's feedback. Concretely, the author agent will review its step-by-step thoughts $t_i$ and incorporate the meta-reviewer's suggestions to identify and correct errors (see Appendix D.1.4 for the prompt template). Once all steps have been re-examined, the

author agent produces an updated answer $y_i^*$. This process can be formulated as:

$$y_i^* = \mathcal{A}(t_i, y_i, m_i) \tag{4}$$

## 3.3 ANALOGY TO RESNET

Residual Networks (ResNets) (He et al., 2016) are a widely adopted deep architecture built from stacked residual blocks connected by skip connections. These shortcuts allow the model to learn a residual function $f(x) = h(x) - x$ rather than the target function $h(x)$ directly, simplifying optimization and making it easier to approximate complex mappings. At the same time, skip connections facilitate gradient backpropagation, mitigating vanishing gradients and enabling more efficient and effective information flow. Together, these properties allow ResNets to train substantially deeper networks and achieve state-of-the-art accuracy across diverse domains.

The comparison between MARS and MAD can be understood through this residual learning perspective. Let $h^*(x)$ denote the ground-truth mapping from a prompt $x$ to its correct answer $y^*$. In MAD, each agent seeks to approximate $h^*(x)$ directly, which requires reasoning from scratch and reconciling outputs with peers, i.e., a process analogous to fitting the full target function in standard networks. By contrast, MARS decomposes the problem: reviewer agents specialize in detecting residual errors between the author's output and the "ideal" answer, while the meta-reviewer integrates these residuals into corrective feedback. The author then updates their response based on this feedback, in the same way that gradients propagate error corrections backward through a ResNet. In this view, MARS shifts the burden from directly modeling the entire solution space to iteratively refining answers via residual error correction, thereby extending the principle of residual learning from neural optimization to system-level multi-agent reasoning.

## 4 EXPERIMENTS

### 4.1 SETUP

**Benchmarks.** We evaluate MARS on three commonly utilized reasoning benchmarking sets: (1) MMLU (Hendrycks et al., 2020) consists of multiple-choice questions across diverse academic subjects such as elementary mathematics, history, computer science, and law; (2) GPQA (Rein et al., 2024) is a graduate-level QA dataset spanning biology, physics, and chemistry, which is of high-quality and even challenging for domain experts, making it a strong test of reasoning ability; (3) GSM8K (Cobbe et al., 2021) is a math word problem dataset that evaluates grade school level mathematical reasoning.

**Models.** We implemented one closed-source and one open-source LLM as agent backbones, including ChatGPT (GPT3.5-turbo) and Mixtral-8×22B (Jiang et al., 2024a). The open-source model is accessed through NVIDIA NIM platform via API calls.

**Baselines.** We compare MARS against several single-agent and multi-agent baselines (detailed experimental settings of the baselines and prompts are available in Appendix C and Appendix D):

- **Chain-of-Thought (CoT).** All models in our experiments are enabled with CoT, which prompts step-by-step reasoning and serves as the most basic baseline.
- **Self-reflection.** This is a single-agent approach in which the model is prompted to evaluate and revise its own response. However, LLMs often struggle to detect their own errors, limiting the effectiveness of this strategy. MARS addresses this limitation by introducing external reviewers who provide independent error detection and correction.
- **Self-consistency.** This approach exploits the inherent stochasticity of LLMs by sampling multiple reasoning paths for the same query and aggregating them through majority voting to produce the final answer.
- **Multi-agent Debate (MAD)** is a framework in which multiple agents engage in a round-table discussion, with each agent generating an answer by combining its own reasoning with the responses of others. All agents play equivalent roles in this process.

**Metrics.** Prior work has primarily assessed reasoning methods by the accuracy of the final answer, where MAD has demonstrated strong performance. In contrast, MARS is designed to improve com-

Table 1: Performance comparison across GPQA, MMLU, and GSM8K for different models and reasoning methods. Reported metrics include: Score: percentage of correct answers, Tokens: average number of tokens generated per query, and Time: average inference time in seconds per query. For each task, the best scores are shown in **bold**, and the second-best scores are underlined.

| Method | GPQA | | | MMLU | | | GSM8K | | |
|---|---|---|---|---|---|---|---|---|---|
| | Score | Tokens | Time | Score | Tokens | Time | Score | Tokens | Time |
| **gpt-3.5-turbo** | | | | | | | | | |
| CoT | 31.00 | 426.69 | 1.83 | 68.00 | 292.65 | 1.20 | 73.33 | 259.82 | 1.34 |
| Self-reflection | 29.67 | 912.92 | 2.61 | 70.33 | 612.04 | 1.80 | 67.00 | 563.22 | 2.16 |
| Self-consistency | 33.67 | 1293.45 | 5.36 | 69.33 | 878.79 | 3.16 | 77.00 | 783.36 | 4.20 |
| MAD | 31.00 | 5042.49 | 11.92 | **71.33** | 3194.74 | 7.64 | **79.00** | 2906.09 | 7.92 |
| MARS | **36.33** | 2479.22 | 6.01 | 71.00 | 1702.21 | 4.71 | 75.67 | 1655.50 | 4.32 |
| **mixtral-8x22b** | | | | | | | | | |
| CoT | 41.33 | 714.70 | 4.09 | 73.33 | 434.57 | 2.23 | 86.33 | 376.23 | 2.36 |
| Self-reflection | 38.00 | 1568.84 | 5.98 | 77.33 | 979.78 | 3.69 | 86.00 | 812.78 | 3.72 |
| Self-consistency | 41.67 | 2144.68 | 12.91 | 76.33 | 1327.86 | 6.76 | 87.00 | 1123.55 | 7.06 |
| MAD | **47.00** | 8796.42 | 21.89 | 76.67 | 4446.02 | 11.15 | 87.00 | 4272.53 | 11.82 |
| MARS | 44.00 | 3894.23 | 9.82 | **77.67** | 2400.47 | 6.40 | **90.33** | 2291.50 | 6.49 |

putational efficiency in LLM-based reasoning, with success defined by achieving accuracy comparable to MAD while substantially reducing resource consumption. Accordingly, we evaluate methods based on the integration of three metrics: (i) final-answer accuracy, (ii) average number of tokens generated per query, and (iii) average inference time (seconds) per query.

## 4.2 MAIN RESULTS

The primary experimental results in all models and datasets are summarized in Table 1. We highlight three key observations:

(1) **Accuracy.** MAD and MARS consistently rank among the top two methods in final-answer accuracy, outperforming all other baselines. Their relative advantage depends on the backbone model and task: for example, on GSM8K with ChatGPT, MAD reaches 79% accuracy versus 75.67% for MARS, whereas on GPQA with the same model, MARS achieves 36.33%, exceeding MAD by more than 5%. Likewise, with Mixtral on GSM8K, MARS attains 90.33%, which is substantially higher than MAD. Overall, MARS achieves comparable accuracy to MAD across diverse models and tasks.

(2) **Token consumptions.** Although both MAD and MARS achieve high accuracy across reasoning tasks, their multi-agent design naturally requires higher token usage compared to single-agent baselines. Notably, MARS reduces the average number of tokens per query by roughly 50% relative to MAD. Given that the two methods achieve comparable accuracy, this demonstrates that MARS makes the multi-agent collaboration paradigm substantially more efficient while preserving response quality.

(3) **Inference time.** Another critical aspect of a reasoning strategy is inference time, particularly for interactive applications such as ChatGPT. As shown in Table 1, MARS reduces inference time by roughly 50% compared to MAD, enabling more responsive user interactions and more scalable large-scale experimentation.

Our results reveal a key limitation of prior MAD-inspired strategies: their gains in reasoning quality come with prohibitive computational overhead. MARS fills this efficiency gap to a certain level, reducing token usage and inference time by about 50% while preserving accuracy. This advance makes multi-agent reasoning not only more effective but also practical for large-scale and real-time applications.

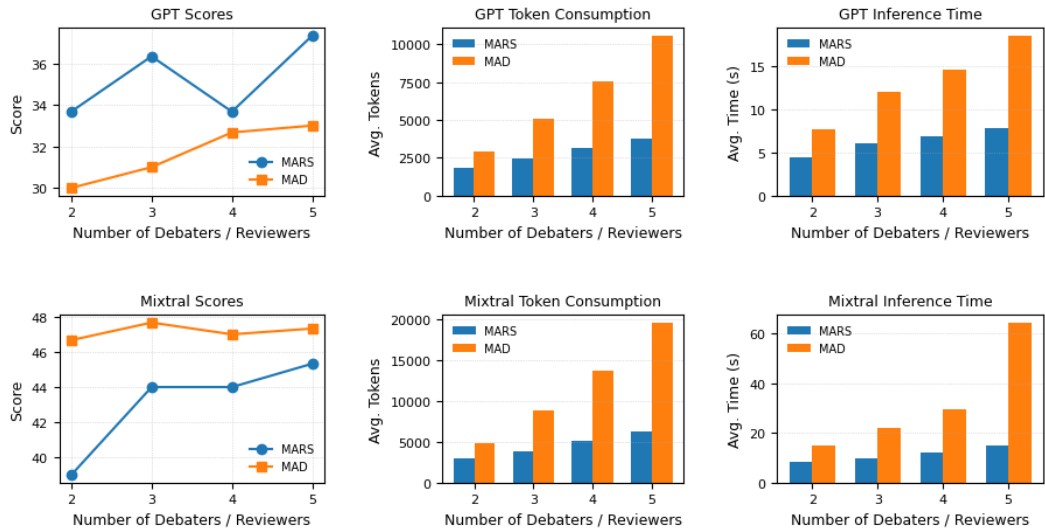

Figure 2: Comparison of MARS and MAD on GPQA with varying number of agents. Row 1: using ChatGPT as the backbone; Row 2: using Mixtral as the backbone. Column 1: accuracy scores; Column 2: averaged number of tokens; Column 3: averaged inference time.

### 4.3 ADDITIONAL ANALYSIS

#### 4.3.1 SCALABILITY

In this section, we evaluate the scalability of MARS versus MAD to further demonstrate the efficiency advantage of MARS. Figure 2 presents their performance across different backbone models and varying numbers of agents (debaters or reviewers, including the meta-reviewer). From the left column of Figure 2, we observe that the accuracy of both methods generally increases as the number of agents grows, consistent with prior findings that collaborative reasoning benefits from larger agent pools. This trend suggests that scaling the number of agents can enhance reasoning performance in multi-agent frameworks. Noted, MAD employs a fully connected communication structure that causes computational cost per query to grow rapidly as the number of agents increases. As shown in the second and third columns of Figure 2, both token usage and inference time grow sharply with additional debaters, severely limiting scalability. In contrast, MARS avoids this bottleneck by designing the parallel reviewer to generate independent evaluations and a meta-reviewer agent to aggregate reviews. This structure enables MARS to improve reasoning efficiency while maintaining performance, with token consumption and inference time growing linearly with the number of reviewers.

#### 4.3.2 EVALUATING DIFFERENT MIXTURES OF MODELS

In this section, we evaluate how heterogeneous backbone models affect the reasoning outcomes. In previous analyses, all agents were driven by the same LLM. Recent work suggests that different models may provide complementary perspectives. Motivated by this, we evaluate diverse combinations of backbone models to test whether reviewer or meta-reviewer heterogeneity affects performance. Table 2 reports the results, from which three key observations emerge: (1) Employing stronger reviewers can improve the quality of the author's output. (2) The overall performance remains bounded by the author's own capability, for instance, even when all reviewers use Mixtral, a ChatGPT author cannot exceed the performance of the Mixtral author. (3) Interestingly, when ChatGPT serves as the reviewers and meta-reviewer, Mixtral surpasses the accuracy achieved under the homogeneous setting where all agents use Mixtral, suggesting that model diversity is a nontrivial factor, which will be worthy of further investigations.

Table 2: Comparison of different author–reviewer–meta-reviewer configurations on GPQA.

| Author | Reviewers | Meta-reviewer | Score | Tokens | Time |
|--------|-----------|---------------|-------|--------|------|
| gpt | gpt+mixtral | gpt | 35.33 | 2830.46 | 7.53 |
| gpt | mixtral+mixtral | gpt | 36.00 | 3120.75 | 7.71 |
| gpt | gpt+mixtral | mixtral | 38.00 | 3104.36 | 8.64 |
| gpt | mixtral+mixtral | mixtral | 39.00 | 3212.75 | 9.14 |
| mixtral | gpt+mixtral | mixtral | 43.50 | 3824.76 | 9.62 |
| mixtral | gpt+gpt | mixtral | 41.67 | 3581.21 | 8.65 |
| mixtral | gpt+mixtral | gpt | 42.23 | 3725.06 | 8.64 |
| mixtral | gpt+gpt | gpt | 46.40 | 3479.96 | 8.14 |

### 4.3.3 EFFECT OF REVIEWER PERSONAS

Recent work suggests that assigning distinct personas to agents can improve performance by promoting diversity. Motivated by this, we explored whether diversifying reviewers through predefined personas could enhance feedback quality and overall reasoning performance. In this variant, namely MARS-P, reviewers were instructed to adopt specific roles (e.g., conservative vs. aggressive; see Appendix D.3 for prompt details) when evaluating the author's response.

Surprisingly, as shown in Figure 3, introducing personas did not yield improvements across tasks. Analysis of the review content suggests two main factors: (i) the inherent randomness of LLMs already provides a degree of diversity in reviewer feedback, and persona prompts did not add significant variance; and (ii) when the author's response is correct, aggressive reviewers often generate unnecessarily critical or noisy feedback, which can confuse the meta-reviewer and cause overcorrections. These results indicate that relying on the model's natural variability might be more effective than enforcing artificial personas.

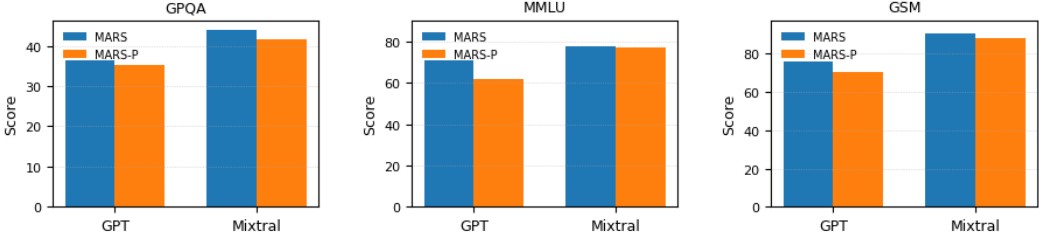

Figure 3: Comparison of MARS with its variant MARS-P, where reviewers are assigned different predefined personas, across multiple tasks. Incorporating personas for reviewers does not yield improvement in reasoning performance.

### 4.4 CASE STUDY

To illustrate how MARS operates, we present a case study from the GSM dataset that demonstrates the framework's step-by-step reasoning process (Figure 4; additional examples appear in Appendix E). The problem describes four dogs, where the first weighs 10 pounds, and each subsequent dog weighs a constant multiple of the previous one. The task is to determine the weight of the fourth dog.

The author agent began by denoting the first dog's weight as $A$ and correctly deriving that the fourth weight should be $22A$. However, it mistakenly recomputed $A$ as $5/11$, which contradicts the given information and led to an incorrect final answer. During the review phase, Reviewer 1 successfully identified this error and explicitly flagged the inconsistency; Reviewer 2 also rejected the author's answer but failed to provide a clear justification. At the meta-review stage, the meta-reviewer inte-

grated these signals, recognized the erroneous recomputation of $A$, and instructed the author to recalculate using the correct initial weight. Incorporating this feedback, the author substituted $A = 10$ pounds and derived the correct result of $22A = 220$ pounds.

We highlight three insights from this case: (1) *Error propagation in long reasoning chains.* For tasks requiring multiple reasoning steps (seven in this example), even a minor mistake can cascade into an incorrect final answer, underscoring the importance of intermediate verifications for the reasoning process. (2) *Reviewer contributions.* Reviewers in MARS can identify errors in the reasoning chain and provide detailed feedback to guide answer revision. While not all reviewers generate high-quality comments, this variability suggests potential benefits from involving multiple reviewers. (3) *Effectiveness of MARS.* By incorporating external feedback rather than relying solely on self-correction, MARS overcomes limitations of single-agent reasoning and improves final accuracy without incurring excessive computational cost.

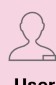

**Problem:** Four dogs sat in a line within the veterinarian's waiting room. The first dog was the Affenpinscher, who weighed only 10 pounds. Next to him sat a Dachshund, who weighed twice as much as the Affenpinscher. Next to the Dachshund sat a Papillon, who weighed one-fourth as much as the Dachshund. And at the end of the line sat a Mastiff, who weighed 44 times the weight of the Papillon. **How much did the Mastiff weigh, in pounds?**

**User**

**Initial response**

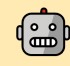

**Author**

Thoughts: 1. Let's denote the weight of the Affenpinscher as A. 2. The Dachshund weighs 2A. 3. The Papillon weighs (1/4)(2A) = 1/2A. 4. The Mastiff weighs 44(1/2A) = 22A. 5. Since the Mastiff's weight is 22A and the given weight of the Affenpinscher is 10 pounds, we can set up the equation 22A = 10. 6. Solving for A, we get A = 10/22 = 5/11 pounds. 7. Finally, the weight of the Mastiff is 22A = 22 * (5/11) = 110/11 = 10 pounds. **Answer: 10 pounds**"

**Review**

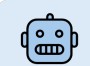

**Reviewer 1**

Decision: Wrong. Confidence: 5. Justification: The author incorrectly calculated the weight of the Affenpinscher as 5/11 pounds, which is not consistent with the given weight of 10 pounds.

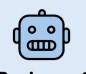

**Reviewer 2**

Decision: wrong. Confidence: 5 Justification: The author's computations are incorrect. While their method seems consistent with the original problem, the calculations lead to errors in the final answer.

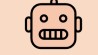

**Meta-reviewer**

Meta-decision: Wrong. Justification: The author made a mistake in their calculations, specifically in determining the weight of the Mastiff. When recalculating based on the correct weight of the Affenpinscher, the answer should be different. Suggestions: Correct the calculations for the weight of the Mastiff.

**Rebuttal**

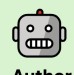

**Author**

Reasons: Accepting the meta-reviewer's suggestions because upon reevaluating the calculations, it seems an error was made in the initial response. Thoughts: 1. The Affenpinscher weighs 10 pounds. Let's denote A as the weight of the Affenpinscher. 2. The Dachshund weighs 2A. 3. The Papillon weighs 1/4 * 2A = 1/2 * A. 4. The Mastiff weighs 44 * (1/2 * A) = 22A. 5. Given that Affenpinscher weighs 10 pounds, A = 10 pounds. 6. Therefore, the weight of the Mastiff is 22A = 22 * 10 = 220 pounds. **Answer: 220 pounds**

Figure 4: Case study of MARS on a GSM example. Upon receiving the user query, the author agent first generates an initial response but incorrectly recomputed a given variable, leading to an incorrect final answer. The reviewers identified the mistake and generated feedback, which guided the author agent to revise the solution, resulting in the correct answer.

## 5 CONCLUSION

In this work, we introduced MARS (Multi-Agent Review System), a framework that enables efficient collaboration among LLMs through a propose–review–feedback–update architecture. MARS addresses the limitations of self-reflection in single-agent settings, enhances reasoning quality, and avoids the heavy computational cost of debate-based methods. Extensive experiments across diverse benchmarks and models show that MARS consistently outperforms strong baselines and matches the accuracy of Multi-Agent Debate while reducing resource consumption by about 50%. These findings suggest that direct communication among all agents is not required for effective collaboration, and that the review-style architectures provide a promising paradigm for scalable and efficient multi-agent reasoning. Limitations and directions for future work are discussed in Appendix A.

## REPRODUCIBILITY STATEMENT

The source code for implementing both MARS and the compared baselines are temporarily available through an anonymous GitHub repository (see link at the end of the abstract), and will be made public after the review process. We also gave a detailed introduction of the prompt templates in Appendix D. All of the datasets used in our experiments are publicly available (see section 4.1 for the references), and are also included in the repository. Users can follow instructions in the README file of the repository to run our model and reproduce results in the experiments.

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

# A    LIMITATIONS

Although MARS achieved strong performance in both accuracy and efficiency, there are still several challenges remaining to be solved. (1) Confidence estimation. The reviewer agents output a confidence score, which helps the meta-reviewer to estimate the reliability of the review comments. In this paper, the confidence score is derived directly from the model's verbalized confidence via prompting. However, recent studies revealed that LLMs tend to be overconfident when using verbalized confidence(Xiong et al., 2023). In our experiments, we did notice that many reviewers tend to give a score of 5 as their confidence level, which may affect the meta-reviewer's judgment. Future work should use more accurate confidence estimation derived by advanced techniques. (2) Over-correction. In MARS, the decision and suggestions of meta-reviewer will highly influence the final answer, especially when the meta-decision is *reject*. One potential issue is that a correct initial response may be overturned if the meta-reviewer provides negative feedback, guiding the author to a wrong direction. Although we attempted to mitigate this by curating the feedback prompt template (the author is asked to insist on its initial answer when disagreeing with the meta-reviewer, see Appendix D.1.4 for the prompt), we cannot guarantee to avoid it in our current setting. Future work should include how to design more effective mechanisms to prevent unnecessary revisions.

# B    MARS ALGORITHM

We summarize the pipeline of MARS (propose-review-feedback-update) as the following algorithm. Details are introduced in section 3.

---

**Algorithm 1:** MARS algorithm

---

**Input:** input query $x$, author model $\mathcal{A}$, $m$ reviewer models $\{\mathcal{R}_j \mid j = 1, \ldots, m\}$,
   meta-reviewer model $\mathcal{M}$
**Output:** final answer $y^*$
**Author Stage:**
$t, y \leftarrow \mathcal{A}(x)$ ;   // Generate thoughts and initial answer (Eq. 1)
**Review Stage:**
**for** $j = 1$ **to** $m$ **do**
 | $r_j \leftarrow \mathcal{R}_j(x, t, y)$ ;  // Each reviewer gives decision, confidence,
 |  justification (Eq. 2)
$r \leftarrow \{r_j \mid j = 1, \ldots, m\}$ ;     // Collecting reviewer comments
**Meta-review Stage:**
$m \leftarrow \mathcal{M}(x, t, y, r)$ ;  // Summarize reviews, resolve conflicts, make
 final decision (Eq. 3)
**Rebuttal Stage:**
**if** $m.decision = accept$ **then**
 | $y^* \leftarrow y$ ;          // Adopt initial answer
**else**
 | $y^* \leftarrow \mathcal{A}(t, y, m)$ ;  // Revise with meta-review feedback (Eq. 4)
**return** $y^*$

---

# C    EXPERIMENT DETAILS

In our experiments, we followed the standard manner of CoT by asking the model to think about the user query step-by-step. For a fair comparison, we adopt a consistent prompt design: all methods use CoT prompting for the initial reasoning process. For example, the initial reasoning step in self-reflection and each reasoning path in self-consistency share the same CoT prompt template. Reflection, debate, review, and feedback methods require distinct prompt styles due to their structural differences. Detailed prompt templates are provided in Appendix D.

We used the standard OpenAI SDK for API calls of all the models used in this work. All of the parameters are set as their default values. For fair comparison, we use the same number of reviewers

(reviewer + meta-reviewer) and debaters in MARS and MAD. Both methods will run one update round after the initial round. While MARS is naturally extendable to multiple rounds of review and revision, we leave a systematic study of this setting to future work due to time and cost constraints.

# D    PROMPT TEMPLATES

In this section, we provide prompt templates for all the methods used in this paper, including both MARS and the compared baselines.

## D.1    PROMPT TEMPLATES FOR MARS

### D.1.1    AUTHOR AGENT

The author agent is prompted to give its initial response using chain-of-thoughts. The following prompt shows that the author receives the user query, generates intermediate thoughts on the problem, and then the final answer. Specifically, the answer format shown here is for QA problems. When applied to math datasets, the format could be changed to

"Answer: [the final numerical answer]"
"Your final answer must be a single numerical number at the end of the response."

```
You are an assistant. Please help to solve the following problem:
{user_query}

Give your thoughts about the question and the final answer in the
    following format:
Thoughts: [your thoughts with immediate results]
Answer: [the final single capital letter answer in the form (X). X is
    chosen from [A,B,C,D]]

Your final answer must be a single capital letter in the form (X)
at the end of the response.
```

### D.1.2    REVIEWER AGENT

```
You are a reviewer. The author has submitted the following answer:

Question: {user_query}
Answer: {author_response}

Please evaluate the correctness of the author's response.
Follow the instructions and format strictly:

Evaluation criteria:
1. Faithfulness: check whether the author's answers and thoughts are
   consistent with known facts.
2. Correctness: check whether each step in the author's answer and
   thoughts is correct.

Your output format must be:

Decision: [right | wrong]
Confidence: [1-5] (5 = highest confidence)
Justification: [reasons or author mistakes supporting your decision]
Answer: [your recommended answer]
```

### D.1.3    META-REVIEWER AGENT

```
You are the meta-reviewer. The author has submitted an answer.

Question: {user_query}
```

```
Answer: {author_response}

You must decide whether the answer is correct based on both your own
knowledge and the reviewers' comments below:

--- Reviewer Feedback ---
{combined_reviews}

Do not only rely on the reviewers, you must also think by yourself.

Provide your conclusion in the following format:

Decision: [right | wrong]
Justification: [reasons of your decision]
Suggestions: [your suggestions for updating the answer, only needed
              when decision is wrong]
Answer: [your recommended answer]
```

### D.1.4 FEEDBACK TO AUTHOR

To avoid over-correction, the author is prompted to think carefully before accepting the meta-reviewer's suggestions. The author will only start to revise its initial answer when it strongly agrees with the meta-reviewer. After the decision is made, it will move on to update its answer following the feedback.

```
Your answer was reviewed and marked as incorrect by the meta-reviewer.

--- Meta-reviewer Feedback ---
{meta_decision}

If you strongly agree with the meta-reviewer's suggestions, revise your
answer accordingly. If you disagree, insist on your initial answer and
repeat it. Do not always trust the meta-reviewer -- you must think for
yourself whether to trust the suggestions.

Make sure to state your reasoning and final answer in this format:

Reasons: [your reasons of accepting or rejecting the suggestions]
Thoughts: [your new step-by-step thoughts on the problem after
           considering the suggestions]
Answer: [the final single capital letter answer in the form (X).
         X is chosen from [A,B,C,D]]
```

## D.2 PROMPT TEMPLATES FOR BASELINES

In this section, we show the prompt templates for the baselines. For a fair comparison, the same CoT template is applied to the initial input of all the techniques. Therefore, we did not include the prompts for self-consistency here since it just applies the same prompt several times. For self-reflection, we provide the prompt for the reflection step. For MAD, we provide both the prompts for the initial response and the debate process for each agent.

### D.2.1 SELF-REFLECTION PROMPTS

```
You wrote the following response to a problem:

Question: {user_query}
Answer: {response}

Carefully review your own answer. Are there any mistakes or thoughts not
    grounded in the given problem or known facts?
- If yes, explain the problems and revise your answer accordingly.
```

```
- If not, confirm and repeat your initial answer.

Your final response must follow this format:
Mistakes (if any):
Answer: [the final single capital letter answer in the form (X).
         X is chosen from [A,B,C,D]]
```

### D.2.2  DEBATE PROMPT

```
You are an assistant. Please help to solve the following problem:
{user_query}

Give your thoughts about the computation steps and the final answer in
    the following format:
Thoughts: [your step-by-step thinking process with immediate results]
Answer: [the final single capital letter answer in the form (X).
         X is chosen from [A,B,C,D]]

Your final answer must be a single capital letter at the end of the
    response.
```

```
These are the solutions to the problem from other agents:

One agent solution: ```{response_1}```
One agent solution: ```{response_2}```
...

Using the solutions from other agents as additional information,
can you provide your final answer to the problem?

Make sure to state your thoughts and new answer with this format:
Thoughts: [your step-by-step thinking process]
Answer: [the final capital letter answer]

Your final answer must be a single capital letter at the end of the
    response.
```

### D.3  REVIEWER PERSONAS

Conservative Persona

```
Your decision should be conservative.

Goal:
Minimize false alarms. Only label 'wrong' when there is a decisive,
demonstrable error that changes the final answer.

Heuristics:
- Require at least TWO independent checks (e.g., recomputation + boundary
    /unit check) before declaring 'wrong'.
- Minor arithmetic slips that do not affect the final answer => prefer '
    right' with notes in Justification.
- Missing small steps != 'wrong' if the step is standard and the result
    follows.

Confidence:
- 5 only if two checks agree
- 4 if one check is strong
- <=3 if any uncertainty remains
```

Aggressive Persona

```
Your decision should be aggressive.

Goal:
Minimize false acceptances. Label 'wrong' when any essential step is
    missing, unjustified, or numerically inconsistent.

Heuristics:
- Penalize ambiguous leaps, missing assumptions/units, and unverified
    claims even if the final number looks plausible.
- If any step is unsupported or contradicts problem constraints, prefer '
    wrong'.
- If correct, still list at least two vulnerabilities the author should
    fix.

Confidence:
- 5 if multiple issues are found or if cross-checks clearly contradict
- 4 when evidence is strong but not exhaustive
- <=3 if partial doubt
```

## E  ADDITIONAL CASE STUDIES

In this section, we present more use cases (see next page) on both math problems from GSM8K and QA problems from GPQA, which further illustrate the workflow of MARS.

## F  LLM USAGE STATEMENT

Large language models (LLMs) were used solely as general-purpose assistive tools for this work. Their usage was limited to grammar checking, language polishing, and LaTeX table formatting. LLMs were not involved in research ideation, methodological design, data analysis, or substantive writing of the manuscript.

| | |
|---|---|
| **Question** | Four friends ordered four pizzas for a total of 64 dollars. If two of the pizzas cost 30 dollars, how much did each of the other two pizzas cost if they cost the same amount? |
| **Author Response** | Thoughts: To solve this problem, we first need to find out the total cost of the two pizzas that are known to cost 30 dollars together. Since there are two pizzas costing 30 dollars, we add their prices to find the total cost of these two pizzas: 30 + 30 = 60 dollars. Next, we need to find out the remaining total cost for the other two pizzas, which means we subtract the total cost of the first two pizzas from the overall total: 64 - 60 = 4 dollars. Since the two remaining pizzas cost the same amount, we divide this remaining total cost by 2 to find the cost per pizza: 4 / 2 = 2 dollars. Therefore, each of the other two pizzas cost 2 dollars. Answer: 2 dollars. |
| **Reviewer 1** | Decision: Wrong
Confidence: 5
Justification: The author's final calculation is incorrect. It was correctly determined that the two known pizzas cost a total of 60 dollars, leaving 4 dollars for the remaining two pizzas. However, the division by 2 is done on the wrong total. The remaining 4 dollars should be divided by the number of remaining pizzas, which is 2. This would give the correct cost per pizza: 4 / 2 = 2 dollars per pizza. But, since the total cost for the other two pizzas is 4 dollars, each of the other two pizzas does not cost 2 dollars. It's the aggregate of both. So, each of the other two pizzas costs 4 dollars / 2 = 2 dollars. Answer: 2 dollars per pizza (each of the other two pizzas costs 2 dollars). |
| **Reviewer 2** | Decision: Wrong
Confidence: 5
Justification: The author's thoughts and calculations are consistent with each other, but they do not match the original problem. The author incorrectly calculates the cost of the two known pizzas as 30 dollars each, which would mean a total of 60 dollars for the two pizzas. However, the original problem states that two pizzas cost a total of 30 dollars, not each. Therefore, the total cost for the two other pizzas should be 64 - 30 = 34 dollars. Since the other two pizzas cost the same amount, we divide this by 2 to find the cost per pizza: 34 / 2 = 17 dollars. Answer: 17 dollars. |
| **Meta-Review** | Decision: Wrong
Justification: Both reviewers have provided incorrect feedback. Reviewer 1 makes a logical error in stating that the 4 dollars should be divided by 2 for the remaining pizzas, but then concludes that each pizza costs 2 dollars, which is the correct answer. Reviewer 2 incorrectly calculates the total cost of the two known pizzas as 60 dollars instead of 30 dollars, leading to an incorrect total for the remaining pizzas and an incorrect final answer. The original solution provided by the author is correct. The two known pizzas cost a total of 30 dollars, leaving 64 - 30 = 34 dollars for the remaining pizzas. Since these two remaining pizzas cost the same amount, we divide this by 2 to find the cost per pizza: 34 / 2 = 17 dollars. However, the author's answer mentions that each of the other two pizzas costs 2 dollars, which is incorrect. The correct answer should be 17 dollars per pizza. Suggestions: The author should correct their final answer to reflect that each of the other two pizzas costs 17 dollars. Answer: 17 dollars. |
| **Author Rebuttal** | Reasons: I agree with the meta-reviewer's suggestions. Upon reviewing, I realized that I made an error in my original computation. The total cost for the remaining pizzas should indeed be 34 dollars, leaving 17 dollars for each of the two remaining pizzas. Thoughts: - The total cost for two known pizzas is 30 dollars. - Subtract the known total from the overall total to find the remaining total: 64 - 30 = 34 dollars. - Since the remaining pizzas cost the same amount, divide this remaining total by 2 to find the cost per pizza: 34 / 2 = 17 dollars. Answer: 17 dollars. |

Table 3: Additional case study showing how MARS corrects an author's mistake through reviews, meta-review, and rebuttal. Example comes from GSM8K.

| Question | Russell works at a pet store and is distributing straw among the rodents. The rats are kept in 3 cages in equal groups and each rat is given 6 pieces of straw. There are 10 cages of hamsters that are kept alone and each hamster is given 5 pieces of straw. There is also a pen of rabbits where 20 pieces of straw are distributed among the rabbits. No straw is used anywhere else in the store. If 160 pieces of straw have been distributed among the small rodents, how many rats are in each cage? |
|---|---|
| Author Response | Thoughts: 1. Let x be the number of rats in each cage. 2. Since there are 3 cages of rats, the total number of rats = 3x. 3. The total number of pieces of straw given to the rats = 3x * 6 = 18x. 4. There are 10 cages of hamsters, so the total number of hamsters = 10. 5. The total number of pieces of straw given to the hamsters = 10 * 5 = 50. 6. The number of rabbits in the pen is unknown, represented as y. 7. The total number of pieces of straw given to the rabbits = 20y. 8. The total number of pieces of straw distributed = 18x + 50 + 20y = 160. Now we can solve for x: 18x + 50 + 20y = 160 18x + 20y = 110 Since we don't have enough information to solve for x, the computation process should stop here. Answer: No final numerical answer. |
| Reviewer 1 | Decision: wrong
Confidence: 5
Justification: The author's computation process is correct up until the final equation dealing with the total pieces of straw distributed. The mistake lies in the setup of the final equation, where the author incorrectly combines the total pieces of straw for rats and rabbits without considering the total number of rabbits and without setting up an equation that connects the number of rats and cages. Answer: The correct way to solve the problem would be to set up a separate equation for the rabbits, considering the total number of rabbits. However, as the problem does not provide the number of rabbits but does provide the total straw distribution, it is not possible to determine the number of rats in each cage. Therefore, the final answer should be stated as "Cannot determine." |
| Reviewer 2 | Decision: wrong
Confidence: 4
Justification: The author's approach overall is correct, but there is one key mistake in their computation. The mistake was in setting up the equation: 18x + 20y = 110 The correct equation should be: 18x + 50 + 20y = 160 After fixing this computation, the correct solution could be reached. Answer: Re-calculate the equation with the correct values. |
| Meta-Review | Decision: wrong
Justification: The author's computation process was mostly correct, but there was a mistake in setting up the final equation. The mistake was not considering the total number of rabbits and not properly combining the total pieces of straw for each type of rodent. Suggestions: Re-calculate the final equation considering the total number of rabbits and properly combining the total pieces of straw for each type of rodent. Answer: Cannot determine. |
| Author Rebuttal | Reasons: I accept the meta-reviewer's suggestions as they pointed out a crucial mistake in considering the total number of rabbits and combining the total pieces of straw for each type of rodent. Thoughts: 1. Let x be the number of rats in each cage. 2. Total number of rats = 3 cages of rats * x rats per cage = 3x rats. 3. Total number of pieces of straw given to the rats = 3x * 6 = 18x pieces. 4. Total number of hamsters = 10 cages of hamsters * 1 hamster per cage = 10 hamsters. 5. Total number of pieces of straw given to the hamsters = 10 * 5 = 50 pieces. 6. Total number of rabbits = 1 pen of rabbits 7. Total number of pieces of straw given to the rabbits = 20 pieces. 8. Total number of pieces of straw distributed = 18x + 50 + 20 = 18x + 70. Given that the total number of pieces of straw distributed among the small rodents is 160: 18x + 70 = 160 18x = 90 x = 90 / 18 x = 5 Answer: Each rat is in a cage with 5 rats. |

Table 4: Additional case study from GSM8K.

| Question | The majority of stars in our Galaxy form and evolve in multi-stellar systems. Below are five potential multi-star systems that are presented. How many of these systems can coexist?
1. W Virginis type star, G2V, M4V, RGB star(1.5Msun)
2. WD (B5 when in the MS) and A0V
3. G2V, K1V, M5V
4. DA4, L4
5. WD (MS mass of 0.85Msun), K3V, A star with a mass of 0.9Msun in the MS.
Options: (A) 2, (B) 4, (C) 3, (D) 1 |
|---|---|
| Author Response | Thoughts: It seems the question is asking about the stability and possibility of certain multi-star systems to coexist. Let's briefly analyze each system:
1. A W Virginis star is a pulsating variable, post main sequence; possible instability.
2. WD (from B5) and A0V: plausible coexistence.
3. G2V, K1V, M5V: stable low-mass system.
4. DA4 and L4: both remnants, stable coexistence.
5. WD (0.85Msun), K3V, A star (0.9Msun): plausible.
From the analysis, all five systems could coexist, but the first may be unstable. Selecting the others gives 4 systems.
Answer: (B) 4 |
| Reviewer 1 | Decision: right
Confidence: 5
Justification: The author's answer is well-reasoned and detailed, correctly identifying possible instability in the first system and concluding that four can coexist. This matches option (B).
Answer: (B) 4 |
| Reviewer 2 | Decision: wrong
Confidence: 5
Justification: The author misinterprets the question. It asks how many systems can coexist, not for a selective stability-based choice. While analysis is good, the answer should reflect the total count. Provided options do not match the author's interpretation.
Answer: (C) 3 |
| Meta-Review | Decision: wrong
Justification: The author's analysis is mostly sound, but they added an interpretation not asked for (selecting only stable systems). The question requires the total number that can coexist. Given the analysis, all except the first system are possible. Thus, the correct answer is 3.
Suggestions: Directly answer the total count rather than reinterpret the question.
Answer: (C) 3 |
| Author Rebuttal | Reasons: I agree with the meta-reviewer's feedback — I misinterpreted the question as requiring a stability-based selection. The correct reading is total number of coexisting systems.
Thoughts: Since all except the first system are possible, the number is 3.
Answer: (C) 3 |

Table 5: Case study on example from GPQA.

