# OpenReview forum: "MARS: Toward More Efficient Multi-agent Collaboration for LLM Reasoning"
_ICLR.cc/2026/Conference — Submitted to ICLR 2026_

### Official Review · Reviewer_9QLU · 2025-10-21

**Soundness:** 3
**Presentation:** 3
**Contribution:** 2
**Rating:** 4
**Confidence:** 4

**Summary:**

This paper constructs a multi-agent collaborative framework that emulates the peer review process, incorporating roles such as author, reviewers, and meta-reviewer. Through this collaborative approach, the framework achieves a significant improvement in efficiency compared to the baseline method MAD, while maintaining comparable accuracy.

**Strengths:**

1. The paper is clear and easy to understand, with a well-defined motivation.
2. This novel collaborative approach demonstrates a certain degree of originality.

**Weaknesses:**

1. The experiments appear somewhat insufficient. Similar works typically conduct extensive evaluations across multiple datasets in diverse domains—such as mathematical reasoning, commonsense reasoning, logical reasoning, and even code generation—to thoroughly demonstrate the generalization capability of the proposed method.

2. The choice of base models is also limited—only GPT and Mixtral were used. The method was not evaluated on the latest and most powerful models such as GPT-5, nor on widely adopted open-source models like Qwen, raising concerns about its generalizability.

3. In real-world peer review processes, reviewers are typically experienced practitioners in the relevant field, and meta-reviewers are often the most senior researchers in the domain. However, in most experiments reported in papers, the underlying models assigned to these roles are usually assumed to possess similar capabilities. How can this setup ensure that meta-reviewers发挥 their full potential to the greatest extent?

4. In Figure 2, why do the results with four reviewers consistently underperform those with three reviewers? I would appreciate some analysis from the authors on this observation.

5. The related work cited by the paper is not comprehensive enough. In fact, since MAD, numerous studies on multi-agent debate have been published, MAD is already considered an outdated approach. Moreover, the set of methods the authors compare against is insufficiently representative, they select only one method from each category for comparison. Moreover, the authors also fail to mention several token-level multi-agent collaboration approaches, such as Determine-Then-Ensemble: Necessity of Top-k Union for Large Language Model Ensembling.

**Questions:**

1. In the main experiments, all participating agents use the same base model, which may lead them to consistently exhibit similar perspectives on the same question. Have the authors analyzed this issue?

2. The performance of some recent, powerful models—such as those in the Qwen-3 series—already exceeds the metrics reported in this paper when used as a single model. Have the authors tried using these models?

3. If, following the meta-reviewer’s suggestion, the authors still fail to provide a correct response during the rebuttal phase, how should such a situation be handled?

---

### Official Review · Reviewer_SH6W · 2025-10-29

**Soundness:** 2
**Presentation:** 3
**Contribution:** 3
**Rating:** 4
**Confidence:** 4

**Summary:**

This paper proposes MARS (Multi-Agent Review System), a framework for improving the efficiency of multi-agent reasoning with LLMs. MARS organizes agents into structured roles, an author, multiple independent reviewers, and a meta-reviewer, to simulate the academic peer-review process. This design maintains reasoning quality while significantly reducing communication overhead. Experiments on MMLU, GPQA, and GSM8K show that MARS achieves accuracy comparable to Multi-Agent Debate (MAD) but reduces token usage and inference time by roughly 50%.

**Strengths:**

1. The paper introduces a novel framework, MARS, which reinterprets multi-agent reasoning as a structured peer-review process.
2. The experiments convincingly demonstrate that MARS can yield improvements in reasoning accuracy and efficiency.

**Weaknesses:**

1. The improvements of MARS are not stable: for instance, GPT-3.5 on MMLU and GSM8K, and Mixtral on GPQA. Without an explanation or statistical validation, it is unclear under what conditions MARS is most beneficial.
2. The ablation primarily uses older/limited backbone LLMs, which weakens the claim that MARS scales under strong contemporary reasoning models.
3. The diversity of both the author model and the reviewer models is critical, a broader range of experiments and more detailed analyses are needed. The impact of model diversity should be systematically investigated to better demonstrate the effectiveness and robustness of the proposed method.
4. Since different models perform differently across tasks, it would be helpful to compare the two models on all datasets under equal computational cost, in order to better understand the upper bound of MARS’s performance (Figure 2).
5. All case studies show very high reviewer confidence scores. It is unclear how low-confidence or conflicting reviews affect meta-review outcomes, or how confidence weighting influences results.
6. The task and dataset diversity needs to be further improved.
7. The effect of multi-round reviews should be explored to demonstrate the scalability and potential extensibility of the proposed framework.

**Questions:**

In Equation (4), should the author model also take x_i as input? It would be helpful to clarify whether including x_i affects the model’s performance or reasoning process.

---

### Official Review · Reviewer_TXW7 · 2025-11-05

**Soundness:** 1
**Presentation:** 2
**Contribution:** 2
**Rating:** 4
**Confidence:** 2

**Summary:**

This paper addresses the high computational cost of Multi-Agent Debate frameworks for LLM reasoning by proposing a novel communication topology MARS inspired by academic peer review. In this method, an "author" agent's solution is evaluated in parallel by independent "reviewers." A "meta-reviewer" then aggregates this feedback to make a final decision or request revisions. The paper claims this hierarchical structure matches the accuracy of MAD while reducing token consumption and inference time by approximately 50%.

**Strengths:**

1.This paper targets to solve high computational and latency costs in MAD frameworks.

2.Compared with MAD, this method reduces token consumption and inference time by ~50% across all tests.

3.MAD's costs scale poorly with more agents, while MARS's costs scale linearly, strongly supporting its efficiency claim.

**Weaknesses:**

1.The core case study provided in Appendix Table 3 demonstrates a completely broken aggregation process. A hallucinating Meta-Reviewer ignoring the single correct Reviewer. This success is only by pure chance.

2.The proposed framework with feedback loop has limited novelty.

3.The ResNet analogy in Sec 3.3 is superficial and provides no real theoretical justification for the method.

4.The central claim of efficiency is highly misleading by exclusively using the extremely expensive MAD framework as its point of comparison. As shown in Table 1, the proposed MARS framework is consistently 2x more expensive in token consumption than the much simpler self-consistency baseline, while MARS's accuracy is often comparable to Self-consistency.

**Questions:**

See weaknesses.

---

### Official Review · Reviewer_Y1w9 · 2025-11-05

**Soundness:** 3
**Presentation:** 3
**Contribution:** 2
**Rating:** 2
**Confidence:** 4

**Summary:**

The paper proposes MARS (Multi-Agent Review System), a role-based alternative to Multi-Agent Debate (MAD) for LLM reasoning. An author produces an initial CoT answer; multiple reviewers independently render decisions with justifications; a meta-reviewer aggregates feedback and either accepts the answer or issues targeted guidance for one rebuttal round by the author. The goal is to preserve MAD-level accuracy while cutting token and time costs; on GPQA/MMLU/GSM8K the authors report roughly ~50% reductions in tokens and latency at comparable accuracy.

**Strengths:**

+ Reviewers operate independently and a single meta-reviewer consolidates feedback, avoiding all-to-all chatter; the paper argues this yields linear growth in tokens/time vs. the steeper growth in MAD as agents increase.

+ On multiple datasets and two backbones, MARS reduces average tokens and inference time by about half relative to MAD at similar accuracy.

+ The appendices provide concrete templates for author, reviewer, meta-reviewer, and feedback stages, aiding reproducibility.

**Weaknesses:**

- The core change vs. MAD is a communication topology and role assignment—useful but incremental. The ResNet analogy is largely descriptive and not validated with theory or targeted ablations demonstrating a “residual-error correction” mechanism.

- Benchmarks are multiple-choice QA (MMLU, GPQA) and grade-school math (GSM8K); there’s no test of robustness (adversarial prompts, noisy questions), no significance testing, and limited analysis of rounds (only one rebuttal studied), which could materially affect both accuracy and cost.

- The GSM8K case study table includes internally inconsistent reasoning between reviewers, meta-review, and author corrections—suggesting the framework can still mis-aggregate noisy signals; this undercuts the claim that meta-review reliably prevents over- or under-correction.

**Questions:**

as in weaknesses

---

### Meta-Review · Area_Chair_zRmb · 2026-01-02

**Summary:**

The recommendation is primarily informed by the insufficient empirical validation and methodological rigor. While MARS shows reduced token usage and inference time compared to MAD, its contribution is incremental, and the results are inconsistent across datasets. The limited model diversity and lack of testing on newer models weaken the claims of generalizability. Additionally, the framework’s handling of conflicting reviews and low-confidence feedback remains unclear. Without stronger validation and broader experiments, the practical utility of MARS remains unproven.

**Reviewer Concerns:**

The rebuttal addressed some empirical concerns, but the critical issue of practical applicability remains unresolved. The study fails to demonstrate how efficiency improvements translate into real-world benefits, and the reliance on a limited set of models restricts the broader relevance of the findings.

**Reviewer Scores:**

Reviewer Y1w9 would likely maintain their score as the unresolved concerns about practical utility and aggregation consistency; Reviewer TXW7 would likely maintain their score as efficiency claims and robustness issues remain unaddressed; Reviewer SH6W would likely maintain their score, as model diversity and low-confidence review handling were not fully resolved; Reviewer 9QLU would likely maintain their score as the inconsistencies and limited dataset diversity undermining generalizability.

---

### Decision · Program_Chairs · 2026-01-26

Reject